# The Presence of Ejaculatory Bulbs in Vasa Deferentia: A Well-Preserved Trait Among Alpheoid Shrimps (Crustacea, Caridea, Alpheoidea)

**DOI:** 10.3390/life15060940

**Published:** 2025-06-11

**Authors:** Lucas Rezende Penido Paschoal, Caio Santos Nogueira, Fernando José Zara

**Affiliations:** 1Invertebrate Morphology Laboratory (IML), Department of Biology, School of Agricultural and Veterinary Studies (FCAV), São Paulo State University (UNESP), São Paulo 14884-900, Jaboticabal, Brazil; lucasrppaschoal@gmail.com (L.R.P.P.);; 2Postgraduate Course in Biological Sciences (Zoology)/Ecology, Evolution and Biodiversity (EcoEvoBio), Biosciences Institute of Rio Claro, São Paulo State University (UNESP), São Paulo 13506-900, Rio Claro, Brazil

**Keywords:** caridean shrimps, ejaculatory duct, histology, reproductive system

## Abstract

The superfamily Alpheoidea comprises eight families: Alpheidae, Barbouriidae, Bythocarididae, Hippolytidae, Lysmatidae, Merguiidae, Ogyrididae and Thoridae. Alpheoids are characterized by possessing two pairs of chelate pereopods, a multiarticulate carpus on pereopod 2, and a narrow strip as the last article on maxilliped 2. However, during the inspection of the reproductive system (RS) of several alpheoids, we consistently observed the presence of ejaculatory bulbs (EBs) in vasa deferentia (VDs) of these shrimps. To investigate whether the presence of EBs in the RS is a conserved trait among Alpheoidea representatives, we analyzed as many species as possible along the Brazilian coast: Alpheidae—5 genera, 19 spp., Hippolytidae—2 genera, 2 spp., Lysmatidae—2 genera, 10 spp., Merguiidae—1 genus, 1 sp., Ogyrididae—1 genus, 2 spp., and Thoridae—1 genus, 1 sp. In addition, we examined representatives of the superfamilies Atyoidea (1 family, 2 genera, 2 spp.), Nematocarcinoidea (1 family, 1 genus, 2 spp.), Palaemonoidea (2 families, 4 genera, 4 spp.) and Processoidea (1 family, 2 genera, 2 spp.) to determine whether EB are present in these groups. Among the groups analyzed, except for the family Alpheidae, most species of alpheoids exhibit an expansion on the ventral portion of the VD in continuity with the lumen of the vas deferens, i.e., the EB. This structure increases the surface area of the VD, consequently increasing the quantity of the seminal material to be ejaculated onto the female. We did not observe the presence of EB in any other of the analyzed superfamilies, suggesting that this structure is exclusive in Alpheoidea. In conclusion, the presence of EB in VD appears to be an exclusive trait in Alpheoidea, being considered a well-preserved synapomorphic trait in this group, except in the family Alpheidae that do not harbor EB, representing a plesiomorphic condition within this superfamily.

## 1. Introduction

Shrimps belonging to the infraorder Caridea are characterized by lamellar gills, a second pleonal pleuron that partially overlaps the first and third pleuron, chelae on their first one or two pairs of pereiopods, females that incubate embryos in their pleon and either extended (>8 stages) or abbreviated (1–3 stages) larval development, as well as direct development in some species. This monophyletic group comprises 13 superfamilies and approximately 40 families. These shrimps can inhabit a wide variety of aquatic environments worldwide, ranging from shallow freshwater environments (e.g., Alpheidae, Euryrhynchidae, and others) to deep marine areas (e.g., Alvinocaridae, Stylodactylidae, and others) [1,2,3,4,5,6].

Within this infraorder, the superfamily Alpheoidea Rafinesque, 1815 stands out as being the most speciose group of caridean shrimps, with 1226 species described according to WoRMS [7]. This superfamily comprises eight families: Alpheidae Rafinesque, 1815 (818 spp.), Barbouriidae Christoffersen, 1987 (10 spp.), Bythocarididae Christoffersen, 1987 (20 spp.), Hippolytidae Spence Bate, 1888 (106 spp.), Lysmatidae Dana, 1852 (57 spp.), Merguiidae Christoffersen, 1990 (2 spp.), Ogyrididae Holthuis, 1955 (13 spp.) and Thoridae Kingsley, 1878 (200 spp.) [7,8]. Most alpheoids are found in shallow marine waters (especially species of Alpheidae shrimps), but they also occur in association with algae, corals or sponges (Hippolytidae and Thoridae), on the roots and limbs of mangroves trees (Merguiidae), in marine caves (Barbouriidae and Lysmatidae), buried in muddy or sandy substrates (Ogyrididae) and even in colder deep waters (>500 m) (Bythocarididae) [4,5,9].

Representatives of this superfamily can be diagnosed by the presence of two pairs of chelate pereopods, a multiarticulate carpus on pereopod 2 (at least in most genera described) and a narrow strip on the terminal half of the mesial margin of the penultimate article as last article on maxilliped 2 [9]. Regarding the sexual systems, gonochorism appears to be less common in this group, being adopted mainly by the most species of Alpheidae and Hippolytidae, and apparently mandatory in Ogyrididae. On the other hand, in the family Merguiidae (all species) and in some Thoridae species (e.g., *Thor amboinensis* (De Man, 1888) and *T. manningi* Chace, 1972), shrimps first mature as males (male phase—MP) and later change to the female sex (female phase—FP) as they grow. In other words, these species are classified as purely protandric hermaphrodites (PH). Representatives of families Barbouriidae and Lysmatidae, as well as some species of Hippolytidae and certain genera within family Alpheidae (e.g., *Athanas* Leach, 1814 and *Salmoneus* Holthuis, 1955), exhibit a sexual system known as protandric simultaneous hermaphroditism (PSH). In this system, shrimps initially reach sexual maturity as MP and, throughout their lives, MP undergo bodily changes during the transitional phase becoming a female which is a simultaneous hermaphrodite (FPSH), capable of mating as well as male or female [4,10,11,12].

Regardless of the type of sexual system adopted by alpheoids, the male reproductive system is arranged in the central portion of the cephalothorax, consisting of a pair of testes (in gonochoric species) or ovotestes (in PH or PSH species, being the lower or caudal portion of this structure being responsible for producing spermatozoa and seminal fluid) associated with the vasa deferentia (VD), which open into the gonopores of the fifth pair of pereiopods [10,11,12,13,14,15,16,17,18,19]. Bauer [14], in his description of mating behavior and spermatophore transfer in the thorid *Heptacarpus sitchensis* (Brandt, 1851), macroscopically identified dilations that occur specifically in the distal region of the VD (DVD), known as ejaculatory bulbs (EBs), and noted: “*A sac-like evagination of the distal portion of the ejaculatory duct, the ejaculatory bulb, is confluent with the lumen of the duct*”. The same author identified these structures in another thorid, *T. manningi*, and described their gradual disappearance during the animal’s sex change [15].

Subsequently, these structures were also observed, either macroscopically and/or microscopically, in other studies involving hippolytid: *Hippolyte niezabitowskii* d’Udekem d’Acoz, 1996 in Manjón-Cabeza et al. [11], and lysmatid shrimps *Lysmata seticaudata* (Risso, 1816) in Charniaux-Cotton [13] and *L. wurdemanni* (Gibbes, 1850) in Bauer and Holt [17], Bortolini and Bauer [18] and Liu et al. [20]. However, none of these authors mentioned the presence of the EB or described these structures in their micrographs or schematic drawings. In contrast, EBs were not observed in alpheid shrimps such as *Synalpheus brevicarpus* (Herrick, 1891) and *S. fritzmuelleri*, Coutière, 1909 analyzed by Moraes et al. [19], nor in *Salmoneus carvachoi* Anker, 2007 analyzed by Oliveira et al. [12].

Surprisingly, during the inspection of the reproductive system of some alpheoid species collected along the Brazilian coast, the presence of EBs in the DVD was consistently observed during specimen dissections (Paschoal; personal communication). Given the recurring presence of these structures in several representatives of Alpheoidea, we conducted macroscopic and microscopic (histological) analyses to determine whether the presence of EBs in VD represents a well-preserved trait within this superfamily. For that, we gathered as much information as possible on the male reproductive system of caridean shrimps from Brazilian aquatic environments. In addition to providing data on Alpheoidea, we also collected, analyzed and compared data from at least two species representing each of the superfamilies Atyoidea De Haan, 1849, Nematocarcinoidea Smith, 1884, Palaemonoidea Rafinesque, 1815, and Processoidea Ortmann, 1896.

## 2. Materials and Methods

### 2.1. Sampling and Species

To obtain the 45 species analyzed in the present study, several types of aquatic environments were sampled across Brazil using different sampling techniques, depending on the type of environment in which the species in question inhabited. All relevant information is summarized in Table 1.

Some species of the genus *Alpheus* Fabricius, 1798, *Typton distinctus* Chace, 1972, the ornamental shrimp *Lysmata ankeri* Rhyne and Lin, 2006 and all *Synalpheus* Spence Bate, 1888 species were obtained throughout active sampling during SCUBA diving at Ilha das Couves and Itaguá beachs, in the municipality of Ubatuba (São Paulo state, southeastern Brazil) during different months of 2022.

Other marine shrimps belonging to the Alpheidae, Hippolytidae, Palaemonidae, Processidae and Thoridae families, as well as some Lysmatidae species, were obtained through nocturnal samplings (22:00–00:00 h) carried out on the rocky shore of Prainha da USP (Ubatuba; 23°30′ 01.7′ S; 45°07′ 07.2′ W) in different months of 2023 and 2024. Specimens were obtained through active collection, where two collectors using fine-mesh aquarium square nets (6 inches) captured the specimens associated with the rocky and sandy substrates during low tide.

The spiny shrimp *Exhippolysmata oplophoroides* (Holthuis, 1948), snapping shrimp *Alpheus intrinsecus* Spence Bate, 1888, the ogyridid *Ogyrides alphaerostris* (Kingsley, 1880) and the processid *Processa hemphilli* Manning and Chace, 1971 were collected by trawling, with a sampling effort of 20–30 min. in daytime and 5 min. at nighttime, using a shrimp fishing boat equipped with double rig nets, also being carried out in the Ubatuba municipality (23º, 27′ S; 45º, 02′ W). All species were collected in different months from 2022 to 2024.

Individuals of semi-terrestrial shrimp *Merguia rhizophorae* (Rathbun, 1900) were obtained through active sampling in mangrove vegetation near to the “Casa de Banhos” at the mouth of the Capibaribe River, Recife municipality, Pernambuco state, northeastern Brazil, during the day in February 2022. The ornamental shrimp *Lysmata dispar* Hayashi, 2007 was collected using artificial refuge structures placed along the rock shores of Recife municipality.

For the amphidromous species, in this case, the atyid *Atya scabra* (Leach, 1816) and the palaemonid *Macrobrachium olfersii* (Wiegmann, 1836), male shrimps were sampled in estuarine environments in Ubatuba municipality with a circular sieve (60 cm in diameter and 1.25 mm mesh), moved several times along the marginal vegetation and macrophyte banks in these environments for 15–30 min by a single collector. Furthermore, hololimnetic palaemonoids *Euryrhynchus amazoniensis* Tiefenbacher, 1978 and *Pseudopalaemon amazonensis* Ramos-Porto, 1979 were obtained through daytime sampling carried out with a circular sieve, moved several times along the marginal vegetation and macrophyte banks in the freshwater environments for 15–30 min by two collectors, in several sections of the Negro River (2°, 56′ S; 60°, 06′ W), located in the municipality of Manaus (Amazonas state, north Brazil) in January 2024.

*Cinetorhynchus erythrostictus* Okuno, 1997 and *C. rigens* (Gordon, 1936) were obtained through a combination of active sampling during SCUBA diving and sampling on the rocky shore (two collectors using aquarium nets) in Taipus de Fora beach located in the municipality of Maraú (Bahia state, northeastern Brazil) in December 2024. The ogyridid *Ogyrides hayi* Williams, 1981 and the alpheid *Potamalpheops tyrymembe* Soledade, Santos and Almeida, 2014 were collected at low tide by using a PVC suction pump (50 mm corer diameter) on a sandy beach in São Sebastião (São Paulo state, southestern Brazil) in September 2024 and on an intertidal mangrove flat in Maraú (Bahia state, northeastern Brazil) in December 2024, respectively. Finally, the malawa shrimp *Caridina pareparensis* De Man, 1892 and the striped cleaner shrimp *Lysmata grabhami* (Gordon, 1935) were bought from different marine aquarium ornamental stores in São Paulo state.

### 2.2. Laboratorial Procedures, Macroscopic and Microscopic Analyses and Histology

After sampling, shrimps were properly transported alive in aerated thermal boxes to laboratory and kept momentarily (24 h) in aquariums with dark bottoms (45 × 25 × 30 cm), with basaltic gravel or sand as substrate and rock fragments to form refuges (temperature 25 ± 0.5 °C), until the time of dissection and processing. These animals were identified through specialized literature for each family and/or specific taxa [9,21,22,23]. After identification, the specimens were measured for carapace length (CL: distance between the posterior margin of the eye orbit and the midpoint of the posterior margin of the carapace) with the aid of an analog caliper (0.02 mm). Subsequently, three to five freshly collected individuals for each species were anesthetized by chilling (−20 °C/5 min), measured and dissected, following the recommendations of Paschoal and Zara [24] and Nogueira et al. [25] for caridean shrimps. Their testes and VD were fixed in 4% paraformaldehyde (in seawater for marine shrimps) for 24 h. After fixation, the male reproductive systems were washed twice in 0.2 M phosphate buffer pH 7.6 (10 min in each bath), dehydrated in an increasing series of ethanol (70 to 95%), and embedded in Leica^®^ historesin glycol-methacrylate. Serial sections of 3.5 to 7 μm thickness were obtained using a rotary microtome. After microtomy, the slides were stained with hematoxylin and eosin (H&E) for a general description of the male reproductive system in Caridea [24,25]. The slides were photographed using a microscope with an imaging system (Leica EC4 camera with Leica^®^ LAS EZ software, version 4.12.0), with the appropriate calibration for the lenses used. The images obtained were analyzed and grouped on photographic plates using Corel Draw 2022^®^ and Adobe Photoshop CS6^®^ software.

### 2.3. Data Analysis

The 45 species were analyzed both macroscopically and microscopically. However, the photographic documentation of some species is not presented here due to the morphologic similarity to other congeners. Nevertheless, we properly evaluated the presence or absence of the EB in the VD of alpheoid, atyoid, nematocarcinoid, palaemonoid and processoid shrimps (Table 1). To achieve this, we obtained the largest possible number of representatives from the Brazilian coast: Alpheidae—5 genera, 19 spp., Hippolytidae—2 genera, 2 spp., Lysmatidae—2 genera, 10 spp., Merguiidae—1 genus, 1 sp., Ogyrididae—1 genus, 2 spp., and Thoridae—1 genus, 1 sp.. We also analyzed species of the superfamilies Atyoidea—1 family, 2 genera, 2 spp., Nematocarcinoidea—1 family, 1 genus, 2 spp., Palaemonoidea—2 families, 4 genera, 4 spp., and Processoidea—1 family, 2 genera, 2 spp. (Table 1). Unfortunately, although there are records of Barbouriidae shrimps in Brazil—*Parhippolyte antiguensis* (Chace, 1972)—we were not able to obtain specimens for comparison. There is no occurrence of Bythocarididae species in Brazil [5].

## 3. Results

### 3.1. Presence/Absence of Ejaculatory Bulbs in Gross Morphology

In general, regardless of the type of sexual system or the superfamily within the infraorder Caridea examined in this study, the vasa deferentia (VDs) are paired organs laterally connected to the testes. Their diameter increases toward the distal region, allowing for the distinction of three regions: proximal (PVD), medial (MVD), and distal (DVD) (*sensu* Oliveira et al. [12], Paschoal and Zara [24] and Nogueira et al. [25]).

The DVD is the region of the VD with the largest caliber (Figure 1, Figure 2 and Figure 3), forming a dilated portion called the ejaculatory duct that ends in the genital pore that opens into the gonopores on the coxopodites of the fifth pair of pereiopods. However, in most Alpheoidea representatives, a recurring structure was observed within the DVD across the most analyzed families: the ejaculatory bulb (EB), an expansion of the VD marked by a fold projected toward the center of the luminal area of the posterior portion of the DVD. The EB is always located on the ventral side of the vas deferens and opens or is continuous with the lumen of the ejaculatory duct (Figure 1).

All alpheoids representatives analyzed from the families Hippolytidae, Lysmatidae, Merguiidae and Thoridae consistently exhibited clear macroscopic evidence of the presence of EBs in the VD, indicated by an evident fold. On the other hand, ogyridid shrimps, do not show obvious macroscopic evidence of a bulb probably due to a high birefringent seminal fluid and discrete fold (Figure 1). In hippolytid shrimps, the EB folds along the entire length of the ejaculatory duct, displaying a tubular aspect, and in *Hippolyte obliquimanus* Dana, 1852, the EB always harbors the androgenic gland (Figure 1A,B). On the other hand, the EBs in lysmatid shrimps exhibited a more globular and even rounded morphology in certain species. This structure can occupy about one-third or even half of the DVD length (Figure 1C–E). The VD of *M. rhizophorae* shows similar morphology to that of *Lysmata jundalini* Rhyne, Calado and dos Santos, 2012, where the EB occupies approximately one-third of the DVD length (Figure 1F). In both species of *Ogyrides* Stebbing, 1914 analyzed here, the EBs were not clearly observed under the gross morphology. Their VD showed a large amount of birefringent seminal fluid in the DVD, but no signs of EBs (Figure 1G). The EB in *T. manningi* presented a pattern very similar to that observed in *H. obliquimanus*, where the EB folds itself in the ejaculatory duct and harbors externally the androgenic gland (Figure 1H).

In contrast, in Alpheidae representatives, the VDs are commonly small, thin, and fragile structures, with few or no birefringent seminal material, especially in the genera *Alpheus* and *Synalpheus*. An exception is *P. tyrymembe*, which exhibits a more complex VD compared to its confamilials (Figure 2). *Potamalpheops tyrymembe* exhibited a deeply coiled DVD, since no EBs fold were observed under histology. The seminal fluid in this region is highly birefringent, and the coiled posterior portion of the DVD—the ejaculatory duct—may lead to a misinterpretation of the presence of EB, creating the false impression of a fold projecting into the lumen (Figure 2A). During our inspection, we found no evidence of EBs in shrimps of the genus *Alpheus* (Figure 2B–E). In the genus *Athanas*, the VD showed a birefringent seminal material in the DVD without signs of EBs (Figure 2F). *Salmoneus ortmanni* (Rankin, 1898) showed a larger ejaculatory duct in DVD when compared to other Alpheidae genera, such as *Alpheus, Athanas,* and *Synalpheus*, with no evidence of EBs (Figure 2G). The VDs of *Synalpheus* shrimps are thin, contain little birefringent seminal material near the genital pore, and lack EBs (Figure 2H).

Representatives of the superfamilies Atyoidea, Nematocarcinoidea, Palaemonoidea and Processoidea analyzed in the present study showed no evidence of harboring EBs in the VD (Figure 3). The VDs of the atyid (Figure 3A), rhynchocinetid (Figure 3B), euryrhynchid (Figure 3C), palaemonid (Figure 3D,E) and processid (Figure 3F) shrimps are tubular in shape commonly surrounded by a thick muscular layer (mainly in the DVD). In the lumen, we observed a large amount of birefringent seminal fluid along the entire length of the VD (Figure 3A–F).

### 3.2. Histological Confirmation and Structure of Ejaculatory Bulbs

The histology of the families Hippolytidae, Lysmatidae, Merguiidae, Ogyrididae, and Thoridae consistently show the presence of EBs in the VD as dilations in the ventral portion of the DVD in continuity with the ejaculatory ducts. The EB is formed by a fold extending toward the luminal center of the VD with its length varying according to species (Figure 4). In hippolytid shrimps, the VD shows a large quantity of spermatozoa surrounded by secretions in their lumen, especially in the DVD. In these shrimps, the EB shows a long fold in the ejaculatory duct and is surrounded by a thick muscular layer. The EB itself harbors a large amount of secretions and spermatozoa (Figure 4A,B). In lysmatid shrimps, the EB fold varies in length, with *L. dispar* exhibiting a fold that almost reaches the end of the DVD (ejaculatory duct). In merguiid shrimps, the EB forms a pronounced fold projecting into the DVD, which is nearly filled with spermatozoa immersed in secretions. In both families, well-developed, elongated or rounded EBs located on the ventral surface of the VD are responsible for accumulating large amounts of spermatozoa and seminal fluid secretions. The EB is easily identifiable by its thick muscular layer, which separates it from the rest of the duct (Figure 4C–E). Contrasting the macroscopic analysis, the VD of the ogyridid shrimps consistently shows distinct EBs with an acute end. In *O. alphaerostris*, the fold of EB, although thin, is relatively long toward the posterior end of DVD and is surrounded by a thick muscular layer connected to a simple epithelium, while the fold of EB in *O. hayi* Williams, 1981 is very short and discrete depicting cuboidal epithelium surrounded by a thin muscular layer. In these species, the EBs accumulate secretions (Figure 4F,G). The VD of *T. manningi* is completely filled with spermatozoa and secretions. The EB in this thorid shrimp shows a simple cubic epithelium fold surrounded by a thick muscular layer and the EB lumen accumulates a large quantity of spermatozoa immersed in secretions (Figure 4H).

Under light microscopy verification, we confirmed the absence of the EBs in the VD for all representatives of the Alpheidae family analyzed here (Figure 5). *Potamalpheops tyrymembe* histological analysis reveals the absence of the EBs, as no inward fold toward the lumen was observed. Instead, the DVD—the ejaculatory duct surrounded by a thick muscular layer—coils upon itself (Figure 5A). This coiling of the ejaculatory duct, along with the thick muscular layer, may lead to a misinterpretation of the presence of an EB during gross anatomical inspection (Figure 2A). The VD in *P. tyrymembe* is completely filled with spermatozoa immersed in a thick layer of seminal fluid secretions, especially observed in the MVD (Figure 5A). In the genera *Alpheus* and *Athanas*, the VD is characterized by having a large number of spermatozoa surrounded by seminal fluid secretions in DVD that show a very wide lumen, particularly in the ejaculatory duct. These secretions provide the presumptive spermatophore layers during the sperm transfer to the female sternum. The VD is formed by a simple epithelium ranging from cuboidal to flattened cells and is surrounded by a thick muscular layer (Figure 5B–D). Curiously, in the analysis of *A.* cf. *packardii*, the individual appeared to have recently copulated—as indicated by an almost empty VD, with few remaining spermatozoa—and no evidence of EBs was detected in its VDs (Figure 5C). The *Synalpheus* shrimps did not show EBs in their VDs. In these shrimps, the DVD is surrounded by a thick muscular layer and contains a small amount of secretion and spermatozoa in its lumen (Figure 5E,F).

No microscopic evidence of EBs in the VDs of representatives of Atyoidea, Nematocarcinoidea, Palaemonoidea, and Processoidea was detected (Figure 6). In atyid (Figure 6A,B), rhynchocinetid (Figure 6C), and palaemonid (Figure 6E,F) shrimps, the VDs are tubular-shaped structures surrounded by a very thick muscular layer connected to a simple epithelium in the DVD, and their lumen contains a large amount of spermatozoa immersed in secretions. In contrast, the VDs of euryrhynchid (Figure 6D) and processid (Figure 6G,H) shrimps show a cuboidal epithelium surrounded by a thin muscular layer, except in the ejaculatory duct, where a thick muscular layer is observed.

## 4. Discussion

Regardless of the type of sexual system adopted by the representatives of the six Alpheoidea families analyzed here, it was possible to verify, through macroscopic and microscopic inspection, an expansion on the ventral portion of the distal region of vas deferens (DVD) in most studied shrimps, known as the ejaculatory bulb (EB). This expansion is marked by a fold projected toward the center of the luminal area of the posterior part of DVD. As demonstrated and suggested by Bauer [14], this structure increases the surface area of the VD and provides additional space for storage. Consequently, such morphological arrangement will allow the male shrimps (or the “male phase” and FPSH, in the case of hermaphrodite species) to increase the amount of seminal fluid forming the spermatophore on the female sternum during mating. Our results support this statement, showing that in most families, the EBs increase the areas of the VD that contain spermatozoa and secretion (see Figure 4, for this statement).

All species of Hippolytidae, Lysmatidae, Merguiidae, Ogyrididae and Thoridae shrimps analyzed in this study harbor EBs in the VDs. However, the observation of the EBs in Ogyrididae representatives was only possible through histological sections. Therefore, we confirm that harboring EBs in VDs is a well-preserved trait within the highly diverse superfamily Alpheoidea. In recent decades, considerable effort has been devoted to explaining and understanding the different types of sexual systems in caridean shrimps, especially the protandric simultaneous hermaphroditism (PSH) [4]. Consequently, several studies have focused on describing the ovotestes and the simultaneous production of female and male gametes in PSH species, but often without addressing the morphology of the DVD [11,16,26,27,28], making it impossible to verify the presence of the EBs in Alpheoidea representatives and to recognize this conserved trait. Although some studies involving lysmatid and thorid shrimps showed the presence of EBs in the DVD of these animals [14,15,17,18,20], no information was available for the other families that comprise the superfamily Alpheoidea. Here, we demonstrated through both macroscopic and microscopic evidence that harboring EBs in the reproductive system is a conserved characteristic within this superfamily. Furthermore, we advocate that the entire reproductive system of caridean shrimps should be properly evaluated, and that the use of histological techniques must be considered mandatory for the accurate characterization of reproductive structures. For instance, the EBs in Ogyrididae representatives were only detected through microscopic analysis; without such inspection, it would have been incorrectly inferred that this family lacks this structure.

The presence of EBs in the VD appears to be uncommon in the family Alpheidae. Only the species *P. tyrymembe* exhibited a deeply coiled DVD, which could lead to a misinterpretation regarding the presence of EBs in this species. Although little is known about the ecology and behavior of *P. tyrymembe*, we suggest that this structural adaptation of the VD likely enhances the efficiency of seminal fluid transfer to the female’s sternal region, particularly in soft muddy substrates. In the field, up to four reproductive males (i.e., with the VD completely filled with spermatozoa and secretion) were observed cohabiting with a single reproductive female (i.e., exhibiting green ovaries filled with mature oocytes) within the burrow of the mangrove crab *Ucides cordatus* (Linnaeus, 1763) (Paschoal, personal communication). Therefore, possessing a greater quantity of seminal fluid may confer an advantage to males that succeed in copulating with the female.

De Grave et al. [8], in their study on the systematic rearrangement of the families Hippolytidae and Barbouriidae, rejected the monophyly of Hippolytidae and supported the recognition of the families Bythocarididae, Lysmatidae, Thoridae, and Merguiidae. Furthermore, these authors demonstrated that these families, together with the family Barbouriidae, form a more derived group in their analyses with the family Alpheidae being considered the most basal group within Alpheoidea. Considering the phylogenetic context and the results obtained in the present study, the absence of EBs in the VDs of Alpheidae representatives may represent a plesiomorphic condition within this superfamily. In contrast, the presence of EBs in the families Hippolytidae, Lysmatidae, Merguiidae, Ogyrididae, and Thoridae can be considered a synapomorphy within Alpheoidea.

When comparing the reproductive system of Alpheoidea shrimps with species from other superfamilies such as Atyoidea, Nematocarcinoidea, Palaemonoidea, and Processoidea, we found that only alpheoid shrimps present EBs in the VD. Therefore, we suggest that this trait may be exclusive to this group. Excluding Alpheoidea, most of the reliable information about the male reproductive system, and consequently the morphology of the VD in caridean shrimps, is concentrated in studies on the genus *Macrobrachium* Spence Bate, 1868 inserted in the family Palaemonidae (Palaemonoidea) [24,25,29,30,31], a few studies involving Atyidae shrimps (Atyoidea) [32,33], and the recent description of the vas deferens in *Ambidexter cochensis* Rodríguez and Lira, 2022 inserted in the family Processidae (Processoidea) [34]. In these shrimps, the VDs are tubular-shaped structures, typically surrounded by a thick muscular layer, with the lumen filled with spermatozoa immersed in secretions. We verified that atyoid, nematocarcinoid, palaemonoid, and processoid shrimps do not show any evidence of harboring EBs in their VD, reinforcing that this trait is unique to alpheoid shrimps. However, both the available material and existing literature suggest that this feature may also occur in Pandaloidea Haworth, 1825, a group closely related to Alpheoidea [1]. For instance, in Hoffman [35], an EB can be observed in the ventral portion of the DVD of *Pandalus platyceros* Brandt, 1851. Similarly, Bergström [36] illustrates the EB in a schematic drawing of the reproductive system of *Pandalus borealis* Krøyer, 1838. Moreover, Okumura et al. [37] provide an image of an EB in the DVD of *Pandalus hypsinotus* Brandt, 1851 during the description of gonadal development and androgenic gland activity during sex change. Nevertheless, Pandaloidea comprises two families, 24 genera, and 203 described species [38]. Since current evidence is limited to three species within the genus *Pandalus* Leach, 1814, inferring that the entire superfamily shares this trait would be premature. Therefore, we emphasize that, to date, only the superfamily Alpheoidea presents this well-defined trait.

## 5. Conclusions

The presence of ejaculatory bulbs in vasa deferentia is a well-preserved trait within the superfamily Alpheoidea, being a synapomorphic characteristic shared by most of the representatives of this superfamily. This structure increases the surface area of the vasa deferentia, allowing males (or the “male phase” and FPHS in hermaphroditic shrimps) to enhance the concentration of seminal material to be ejaculated onto the female. Our results suggest that the presence of EBs can serve as a useful phylogenetic characteristic and may contribute to a better understanding of the evolution of the male reproductive system in Caridea. Furthermore, based on the evidence obtained here, we stress that the entire reproductive system must be thoroughly evaluated and subjected to rigorous macroscopic and microscopic analyses.

## Figures and Tables

**Figure 1 life-15-00940-f001:**
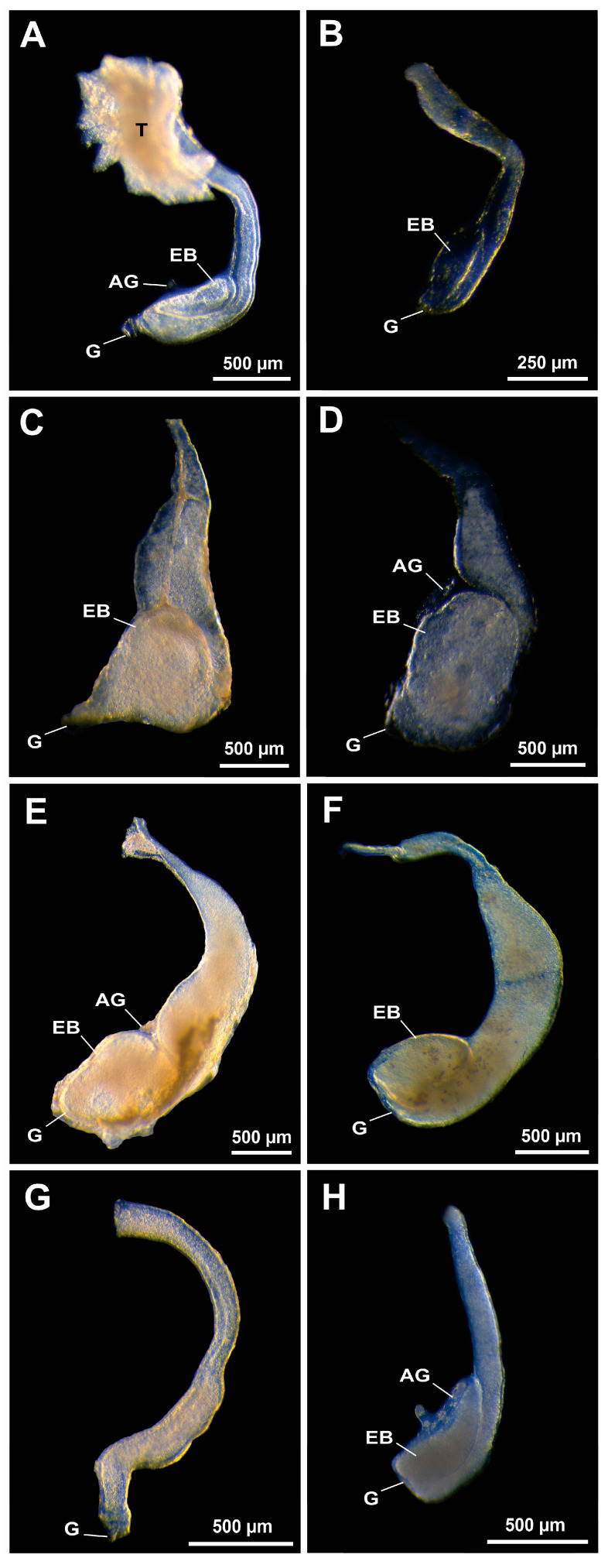
Vas deferens in Hippolytidae (**A**,**B**), Lysmatidae (**C**–**E**), Merguiidae (**F**), Ogyrididae (**G**) and Thoridae (**H**) representatives. (**A**) *Hippolyte obliquimanus* Dana, 1852; (**B**) *Latreutes baueri* Terossi, Almeida and Mantelatto, 2019; (**C**) *Lysmata jundalini* Rhyne, Calado and dos Santos, 2012; (**D**) *Lysmata rauli* Laubenheimer and Rhyne, 2010; (**E**) *Lysmata uncicornis* Holthuis and Maurin, 1952; (**F**) *Merguia rhizophorae* (Rathbun, 1900); (**G**) *Ogyrides hayi* Williams, 1981; (**H**) *Thor manningi* Chace, 1972. AG: androgenic gland; EB: ejaculatory bulb; G: genital pore, T: testes.

**Figure 2 life-15-00940-f002:**
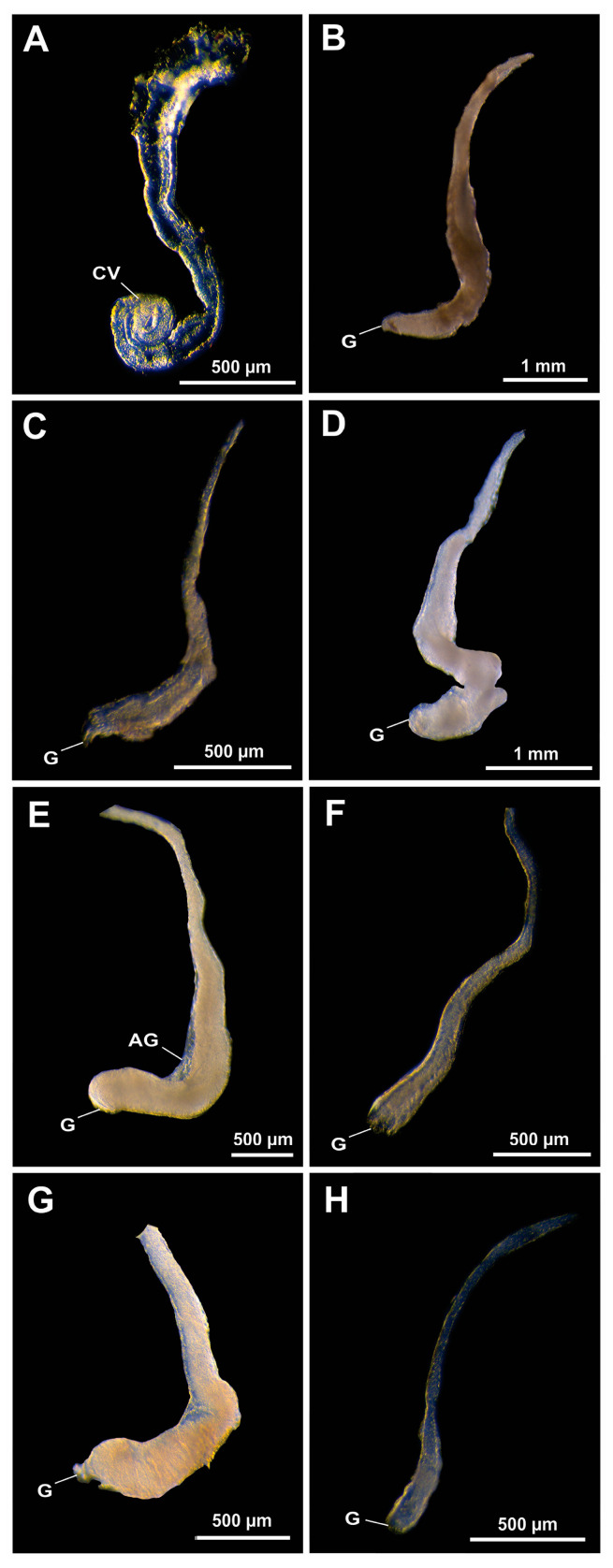
Vas deferens in Alpheidae representatives. (**A**) *Potamalpheops tyrymembe* Soledade, Santos and Almeida, 2014; (**B**) *Alpheus brasileiro* Anker, 2012; (**C**) *Alpheus estuariensis* Christoffersen, 1984; (**D**) *Alpheus intrinsecus* Spence Bate, 1888; (**E**) *Alpheus* cf. *packardii* Kingsley, 1880; (**F**) *Athanas dimorphus* Ortmann, 1894; (**G**) *Salmoneus ortmanni* (Rankin, 1898); (**H**) *Synalpheus fritzmuelleri* Coutière, 1909. AG: androgenic gland; CV: convoluted vas; G: genital pore.

**Figure 3 life-15-00940-f003:**
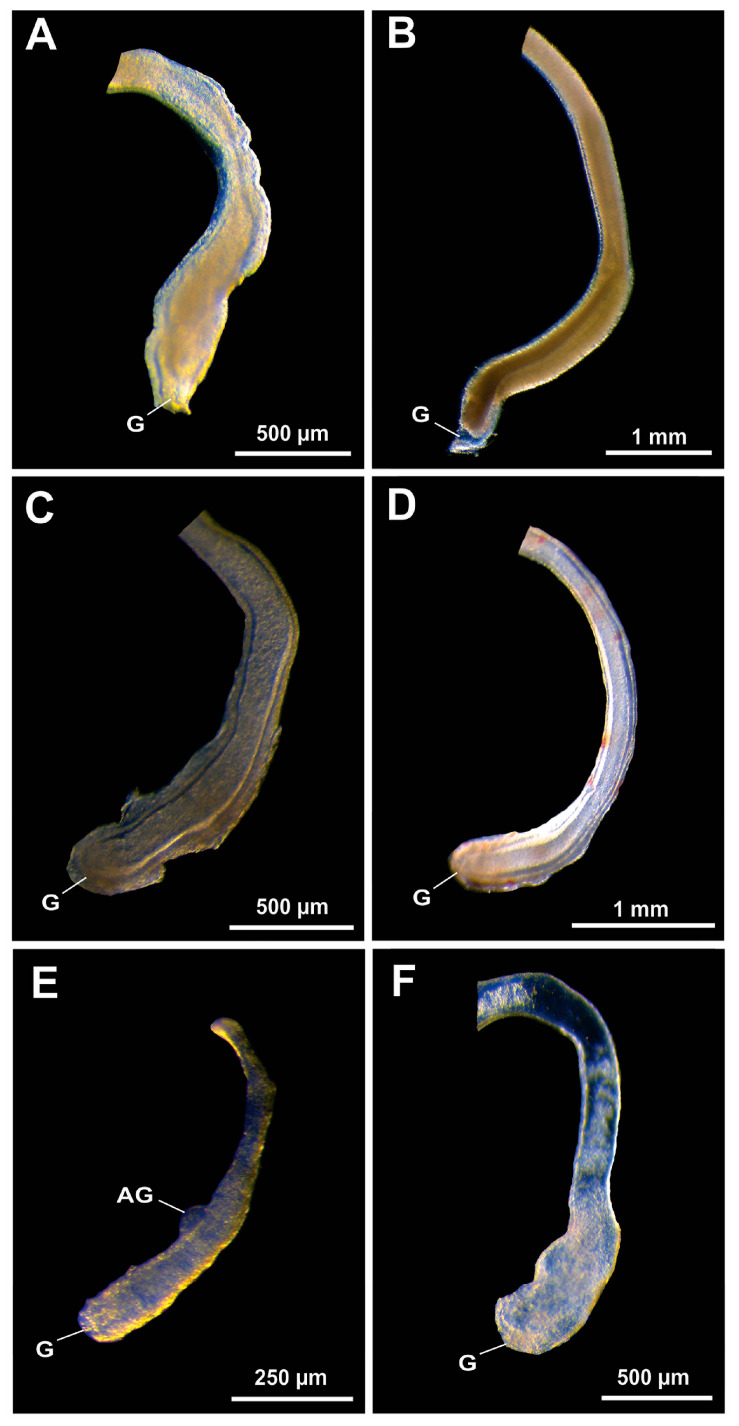
Vas deferens in Atyidae (**A**), Rhynchocinetidae (**B**), Euryrhynchidae (**C**), Palaemonidae (**D**,**E**) and Processidae (**F**) representatives. (**A**) *Caridina pareparensis* De Man, 1892; (**B**) *Cinetorhynchus rigens* (Gordon, 1936); (**C**) *Euryrhynchus amazoniensis* Tiefenbacher, 1978; (**D**) *Pseudopalaemon amazonensis* Ramos-Porto, 1979; (**E**) *Typton distinctus* Chace, 1972; (**F**) *Ambidexter cochensis* Rodríguez and Lira, 2022. AG: androgenic gland; G: genital pore.

**Figure 4 life-15-00940-f004:**
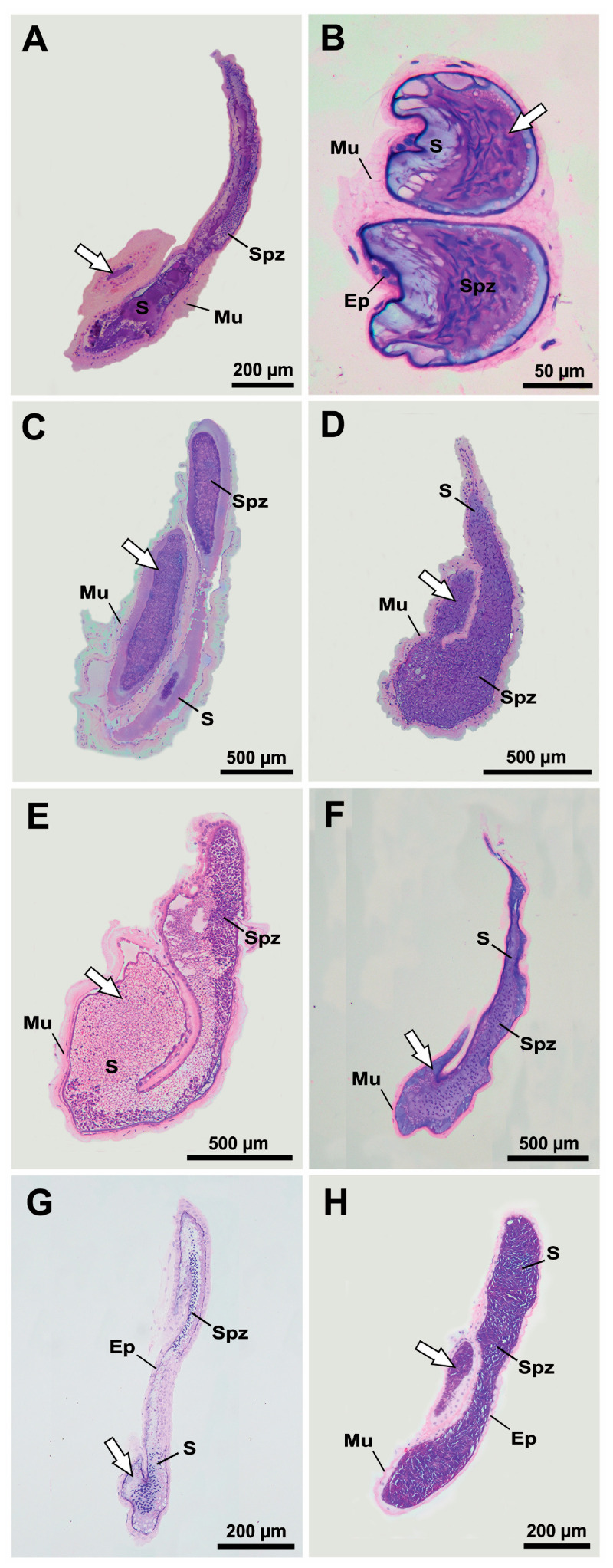
Vas deferens in Hippolytidae (**A**,**B**), Lysmatidae (**C**,**D**), Merguiidae (**E**), Ogyrididae (**F**,**G**) and Thoridae (**H**) representatives. (**A**) *Hippolyte obliquimanus* Dana, 1852; (**B**) *Latreutes baueri* Terossi, Almeida and Mantelatto, 2019; (**C**) *Lysmata dispar* Hayashi, 2007; (**D**) *Lysmata grabhami* (Gordon, 1935); (**E**) *Merguia rhizophorae* (Rathbun, 1900); (**F**) *Ogyrides alphaerostris* (Kingsley, 1880); (**G**) *Ogyrides hayi* Williams, 1981; (**H**) *Thor manningi* Chace, 1972. The white arrows indicate the ejaculatory bulbs. Ep: epithelium; Mu: muscle layer; S: secretion; Spz: spermatozoa.

**Figure 5 life-15-00940-f005:**
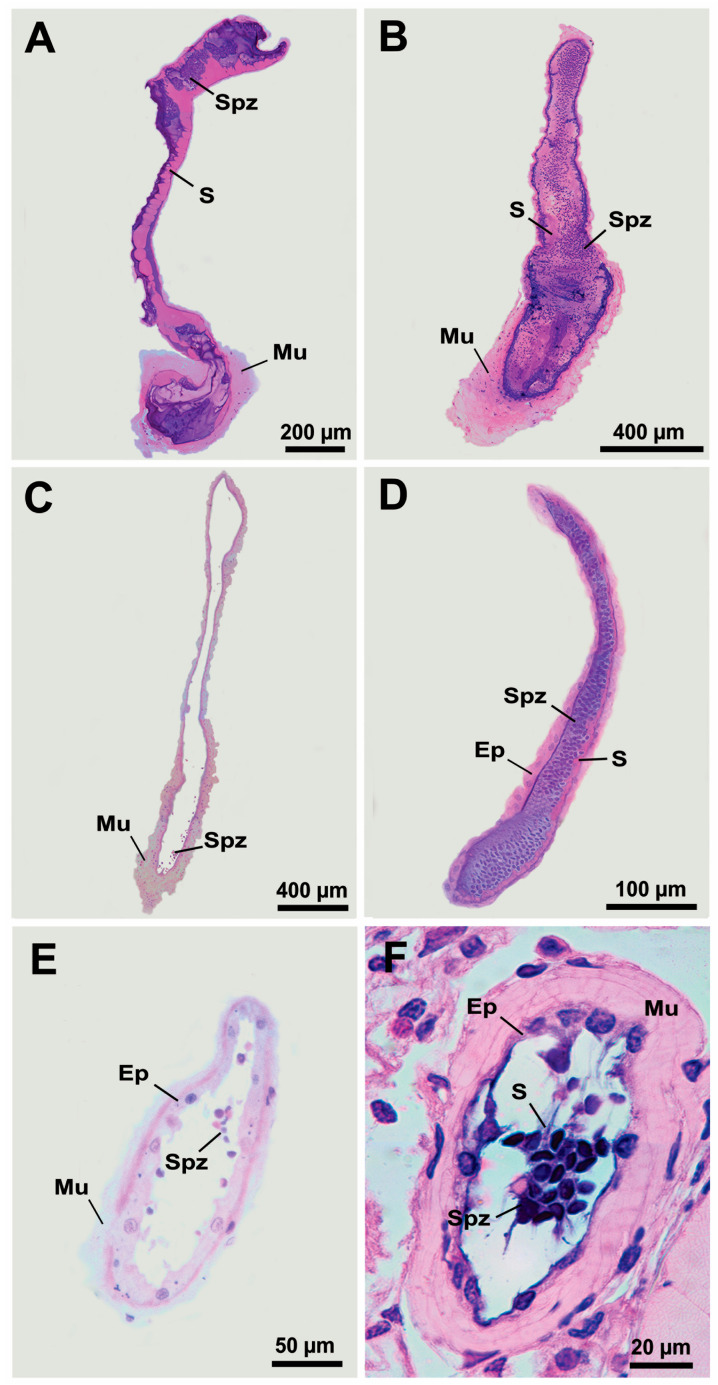
Vas deferens in Alpheidae representatives. (**A**) *Potamalpheops tyrymembe* Soledade, Santos and Almeida, 2014; (**B**) *Alpheus nuttingi* (Schmitt, 1924); (**C**) *Alpheus* cf. *packardii* Kingsley, 1880; (**D**) *Athanas nitescens* (Leach, 1814); (**E**) *Synalpheus brevicarpus* (Herrick, 1891); (**F**) *Synalpheus minus* (Say, 1818). Ep: epithelium; Mu: muscle layer; S: secretion; Spz: spermatozoa.

**Figure 6 life-15-00940-f006:**
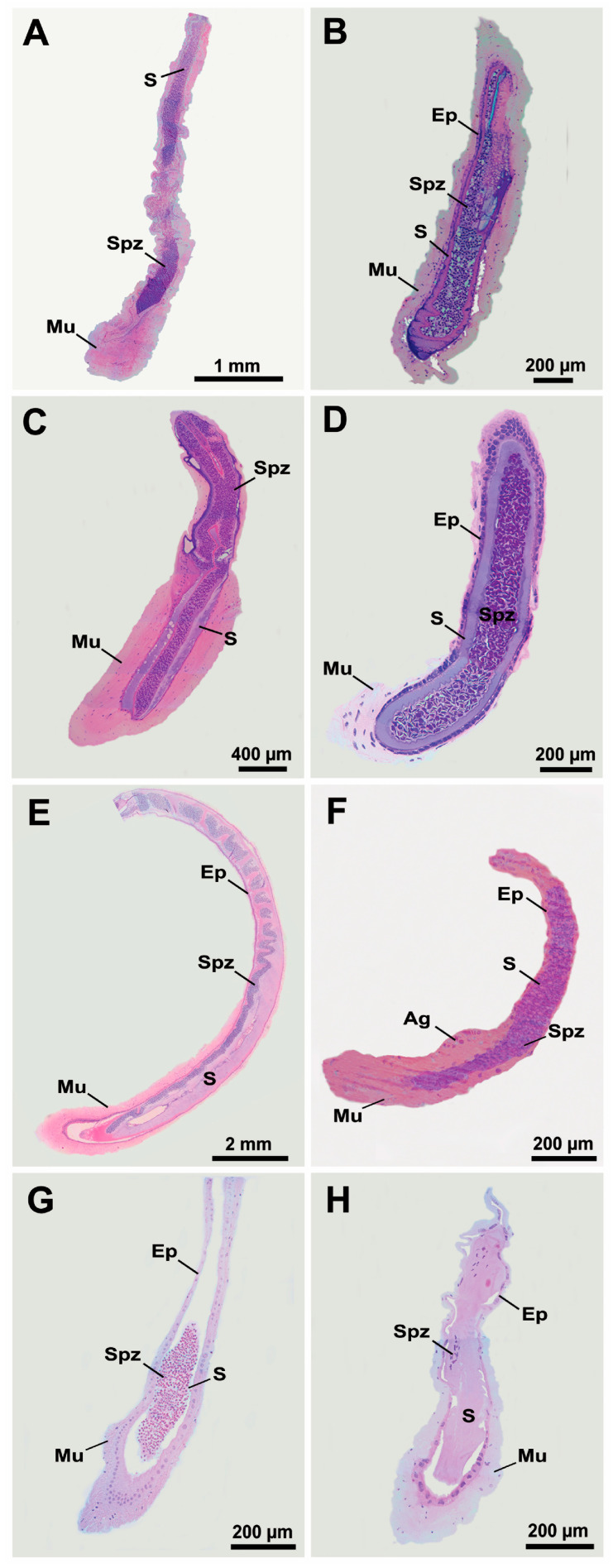
Vas deferens in Atyidae (**A**,**B**), Rhynchocinetidae (**C**), Euryrhynchidae (**D**), Palaemonidae (**E**,**F**) and Processidae (**G**,**H**) representatives. (**A**) *Atya scabra* (Leach, 1816); (**B**) *Caridina pareparensis* De Man, 1892; (**C**) *Cinetorhynchus erythrostictus* Okuno, 1997; (**D**) *Euryrhynchus amazoniensis* Tiefenbacher, 1978; (**E**) *Macrobrachium olfersii* (Wiegmann, 1836); (**F**) *Typton distinctus* Chace, 1972; (**G**) *Ambidexter cochensis* Rodríguez and Lira, 2022; (**H**) *Processa hemphilli* Manning and Chace, 1971. AG: androgenic gland; Ep: epithelium; Mu: muscle layer; S: secretion; Spz: spermatozoa.

**Table 1 life-15-00940-t001:** Caridean shrimps analyzed in the present study, including the sampling methods used to obtaining the specimens, the types of environments where the animals were collected, and the region and state in Brazil where the sampling was conducted. Macro and micro, refer to macroscopic or microscopic data, respectively, of the species presented on the plates in the present study. EB: ejaculatory bulb.

Superfamily	Family	Species	Sampling Methods	Environments	Municipality (State)	Macro	Micro	Presence of EB
Alpheoidea Rafinesque, 1815	Alpheidae Rafinesque, 1815	*Alpheus armillatus* H. Milne Edwards, 1837	manual (hand net)	rocky shore	Ubatuba (São Paulo)	No	No	Absent
		*Alpheus brasileiro* Anker, 2012	manual (hand net)	rocky shore	Ubatuba (São Paulo)	Yes	No	Absent
		*Alpheus buckupi* Almeida, Terossi, Araújo-Silva and Mantelatto, 2013	manual (hand net)	rocky shore	Ubatuba (São Paulo)	No	No	Absent
		*Alpheus carlae* Anker, 2012	manual (hand net)	rocky shore	Ubatuba (São Paulo)	No	No	Absent
		*Alpheus estuariensis* Christoffersen, 1984	manual (hand net)	rocky shore	Ubatuba (São Paulo)	Yes	No	Absent
		*Alpheus formosus* Gibbes, 1850	manual (hand net)	rocky shore	Ubatuba (São Paulo)	No	No	Absent
		*Alpheus intrinsecus* Spence Bate, 1888	trawling (daytime)	marine (in shore)	Ubatuba (São Paulo)	Yes	No	Absent
		*Alpheus nuttingi* (Schmitt, 1924)	manual (hand net)	rocky shore	Ubatuba (São Paulo)	No	Yes	Absent
		*Alpheus* cf. *packardii* Kingsley, 1880	manual (hand net)	rocky shore	Ubatuba (São Paulo)	Yes	Yes	Absent
		*Alpheus thomasi* Hendrix and Gore, 1973	manual (hand net)	rocky shore	Ubatuba (São Paulo)	No	No	Absent
		*Athanas dimorphus* Ortmann, 1894	manual (hand net)	rocky shore	Ubatuba (São Paulo)	Yes	No	Absent
		*Athanas nitescens* (Leach, 1814)	manual (hand net)	rocky shore	Ubatuba (São Paulo)	No	Yes	Absent
		*Potamalpheops tyrymembe* Soledade, Santos and Almeida, 2014	suction pump	mangrove	Maraú (Bahia)	Yes	Yes	Absent
		*Salmoneus ortmanni* (Rankin, 1898)	manual (hand net)	marine (in shore)	São Sebastião (São Paulo)	Yes	No	Absent
		*Synalpheus apioceros* Coutière, 1909	scuba diving	marine (in shore)	Ubatuba (São Paulo)	No	No	Absent
		*Synalpheus brevicarpus* (Herrick, 1891)	scuba diving	marine (in shore)	Ubatuba (São Paulo)	No	Yes	Absent
		*Synalpheus fritzmuelleri* Coutière, 1909	scuba diving	marine (in shore)	Ubatuba (São Paulo)	Yes	No	Absent
		*Synalpheus minus* (Say, 1818)	scuba diving	marine (in shore)	Ubatuba (São Paulo)	No	Yes	Absent
		*Synalpheus ubatuba* Mantelatto, França, Cunha and Almeida, 2023	scuba diving	marine (in shore)	Ubatuba (São Paulo)	No	No	Absent
	Hippolytidae Spence Bate, 1888	*Hippolyte obliquimanus* Dana, 1852	manual (hand net)	rocky shore	Ubatuba (São Paulo)	Yes	Yes	Present
		*Latreutes baueri* Terossi, Almeida and Mantelatto, 2019	manual (hand net)	rocky shore	Ubatuba (São Paulo)	Yes	Yes	Present
	Lysmatidae Dana, 1852	*Exhippolysmata oplophoroides* (Holthuis, 1948)	trawling (daytime)	marine (in shore)	Ubatuba (São Paulo)	No	No	Present
		*Lysmata ankeri* Rhyne and Lin, 2006	scuba diving	marine (in shore)	Ubatuba (São Paulo)	No	No	Present
		*Lysmata bahia* Rhyne and Lin, 2006	manual (hand net)	rocky shore	Ubatuba (São Paulo)	No	No	Present
		*Lysmata dispar* Hayashi, 2007	artificial refuge	rocky shore	Recife (Pernambuco)	No	Yes	Present
		*Lysmata grabhami* (Gordon, 1935)	Aquarism	-	-	No	Yes	Present
		*Lysmata intermedia* (Kingsley, 1878)	manual (hand net)	rocky shore	Ubatuba (São Paulo)	No	No	Present
		*Lysmata jundalini* Rhyne, Calado and dos Santos, 2012	manual (hand net)	rocky shore	Ubatuba (São Paulo)	Yes	No	Present
		*Lysmata rauli* Laubenheimer and Rhyne, 2010	manual (hand net)	rocky shore	Ubatuba (São Paulo)	Yes	No	Present
		*Lysmata uncicornis* Holthuis and Maurin, 1952	manual (hand net)	rocky shore	Ubatuba (São Paulo)	Yes	No	Present
		*Lysmata wurdemanni* (Gibbes, 1850)	manual (hand net)	rocky shore	Ubatuba (São Paulo)	No	No	Present
	Merguiidae Christoffersen, 1990	*Merguia rhizophorae* (Rathbun, 1900)	manual (hand net)	mangrove	Recife (Pernambuco)	Yes	Yes	Present
	Ogyrididae Holthuis, 1955	*Ogyrides alphaerostris* (Kingsley, 1880)	trawling (nighttime)	marine (in shore)	Ubatuba (São Paulo)	No	Yes	Present
		*Ogyrides hayi* Williams, 1981	suction pump	sand beach	São Sebastião (São Paulo)	Yes	Yes	Present
	Thoridae Kingsley, 1878	*Thor manningi* Chace, 1972	hand net	rocky shore	Ubatuba (São Paulo)	Yes	Yes	Present
Atyoidea De Haan, 1849	Atyidae De Haan, 1849	*Atya scabra* (Leach, 1816)	manual (sieve)	river	Ubatuba (São Paulo)	No	Yes	Absent
		*Caridina pareparensis* De Man, 1892	aquarism	-	-	Yes	Yes	Absent
Nematocarcinoidea Smith, 1884	Rhynchocinetidae Ortmann, 1890	*Cinetorhynchus erythrostictus* Okuno, 1997	scuba diving/ manual (hand net)	rocky shore	Maraú (Bahia)	No	Yes	Absent
		*Cinetorhynchus rigens* (Gordon, 1936)	scuba diving/ manual (hand net)	rocky shore	Maraú (Bahia)	Yes	No	Absent
Palaemonoidea Rafinesque, 1815	Euryrhynchidae Holthuis, 1950	*Euryrhynchus amazoniensis* Tiefenbacher, 1978	manual (sieve)	river	Manaus (Amazonas)	Yes	Yes	Absent
	Palaemonidae Rafinesque, 1815	*Macrobrachium olfersii* (Wiegmann, 1836)	manual (sieve)	river	Ubatuba (São Paulo)	No	Yes	Absent
		*Pseudopalaemon amazonensis* Ramos-Porto, 1979	manual (sieve)	river	Manaus (Amazonas)	Yes	No	Absent
		*Typton distinctus* Chace, 1972	scuba diving	marine (in shore)	Ubatuba (São Paulo)	Yes	Yes	Absent
Processoidea Ortmann, 1896	Processidae Ortmann, 1896	*Ambidexter cochensis* Rodríguez and Lira, 2022	hand net	rocky shore	Ubatuba (São Paulo)	Yes	Yes	Absent
		*Processa hemphilli* Manning and Chace, 1971	trawling (nighttime)	Marine (in shore)	Ubatuba (São Paulo)	No	Yes	Absent

## Data Availability

All data are included in this manuscript.

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
