# Peer review of "The Presence of Ejaculatory Bulbs in Vasa Deferentia: A Well-Preserved Trait Among Alpheoid Shrimps (Crustacea, Caridea, Alpheoidea)"

_life, 2025, doi:10.3390/life15060940_

Round 1

Reviewer 1 Report

Comments and Suggestions for Authors

This is an excellent manuscript and interesting and valuable contribution to the knowledge and literature about shrimp sexual biology. I would like the authors to put a sentence at the end of  Abstract about the phylogenetic implication of their work, i..e, the EB is a synapomorphic trait evolved within the Alpheoidea, i.e., not present in the basal Alpheidae, which have the plesiomorphic condition (no EB)  but present in the families derived from it. The SEM and histology are excellent and necessary to support the text.

I have made a few comments and suggestions via sticky notes on the attached pdf that I hope the authors will respond to and which I think will polish a fine study and publication

Author Response

Dear reviewer, we are very grateful for your comments and suggestions. We would like to inform you that all your requests have been addressed. Please see the attached letter, in which all your comments are answered point by point.

Reviewer 2 Report

Comments and Suggestions for Authors

This study investigates the presence of ejaculatory bulbs (EBs) in the vasa deferentia of caridean shrimps, focusing on the superfamily Alpheoidea and comparative groups. Through macroscopic and histological analyses of 45 species from Brazilian coastal habitats, the authors demonstrate that EBs are a conserved, synapomorphic trait in most Alpheoidea families,but notably absent in Alpheidae. The absence of EBs in non-alpheoid superfamilies (Atyoidea, Nematocarcinoidea, Palaemonoidea, Processoidea) supports the exclusivity of this structure to Alpheoidea. This work provides crucial morphological evidence for the systematics of Alpheoidea, identifying EBs as a novel phylogenetic character while highlighting the necessity of combined macro- and microscopic approaches for accurate trait characterization in crustacean reproductive biology. I have reviewed this manuscript thoroughly and find the methodology robust and the conclusions well-supported by the data; I recommend ​​acceptance after minor revisions​​.
​​Specific Revisions Required:​​
1. ​​The "Materials and Methods" section requires reorganization into clearly labeled subsections (e.g., 2.1. Sample Collection and Species, 2.2. Dissection and Macroscopic Analysis, 2.3. Histological Processing, 2.4. Data Analysis) to improve readability and logical flow. The current monolithic paragraph format is difficult to follow.
2. The claim in Results ("Their diameter increases... (data not shown)") requires explicit support. Either include representative images/figures illustrating the PVD, MVD, and DVD regions across key species, or cite specific previous literature establishing this anatomical convention in caridean shrimps. Omitting evidence weakens this foundational anatomical description.
3. ​​The current subheadings "3.1. Macroscopic analysis" and "3.2. Histological analysis" describe methods of observation, not the results obtained. Revise these to reflect the key findings revealed by these methods (e.g., 3.1. Presence/Absence of Ejaculatory Bulbs in Gross Morphology, 3.2. Histological Confirmation and Structure of Ejaculatory Bulbs).

Author Response

(The authors gave the same response as above.)
